# Breaking barriers in crosslinking mass spectrometry with enhanced throughput and sensitivity using Orbitrap Astral

Fränze Müller [1] ✉, Micha J. Birklbauer [2,3], Julia Bubis[1], Karel Stejskal[4], Viktoria Dorfer [2] ✉ & Karl Mechtler [1,4,5] ✉

Crosslinking mass spectrometry is an essential tool for probing protein-protein interactions and structural organization. We here compare Orbitrap Astral and Orbitrap Eclipse instruments using Cas9 crosslinked with PhoX and DSSO under standardized chromatographic and acquisition conditions. The Astral identifies over 40% more unique residue pairs, largely due to increased MS1 sensitivity and efficient detection of low-abundance precursors. Implementation of high-field asymmetric ion mobility spectrometry further increases identifications by 30% through improved precursor filtering. On the Astral, single higher-energy collisional dissociation consistently outperforms stepped fragmentation, particularly at low sample amounts, whereas the Eclipse shows minimal dependence on fragmentation strategy. Gradient optimization experiments demonstrate that longer separations enhance identifications in purified crosslinked samples, while gains plateau in complex backgrounds, indicating the need for enrichment or isolation strategies. Column comparisons show that pore size and particle diameter affect separation efficiency, with the Aurora Ultimate column yielding sharper peaks and more crosslink identifications than PepMap. Together, these findings emphasize that instrument choice, fragmentation mode, and chromatographic design directly influence crosslinking performance. The Astral's combination of sensitivity and scan speed supports comprehensive detection of low-abundance crosslinks, providing deeper structural coverage of protein interaction networks.

Crosslinking mass spectrometry (CLMS) has become a vital tool for studying protein-protein interactions and the three-dimensional architecture of biological systems. By chemically linking interacting residues and analyzing these crosslinked peptides, CLMS complements traditional techniques like cryo-electron microscopy and X-ray crystallography, providing unique insights into protein structures and interactions[1–8]. These complementary insights are invaluable for understanding the intricate interactions underlying cellular machinery and for constructing more comprehensive models of protein complexes[9–12]. Recent advances in CLMS have focused on enhancing crosslinking chemistry, instrumentation, and data analysis. New crosslinkers, including photoactivatable and

[1]Research Institute of Molecular Pathology (IMP), Vienna BioCenter (VBC), Vienna, Austria. [2]Bioinformatics Research Group, University of Applied Sciences Upper Austria, Hagenberg, Upper Austria, Austria. [3]Institute for Symbolic Artificial Intelligence, Johannes Kepler University Linz, Linz, Upper Austria, Austria. [4]Institute of Molecular Biotechnology (IMBA), Vienna BioCenter (VBC), Vienna, Austria. [5]Gregor Mendel Institute of Molecular Plant Biology (GMI), Vienna BioCenter (VBC), Vienna, Austria. ✉e-mail: fraenze.mueller@imp.ac.at; viktoria.dorfer@fh-hagenberg.at; karl.mechtler@imp.ac.at

**Fig. 1 | Overview of the experiment design.** QC samples Cas9 crosslinked with PhoX or DSSO were injected on Astral or Eclipse instruments in parallel with single or stepped higher-energy collisional dissociation (HCD) methods to compare the performance of the instruments and fragmentation methods. Proteome Discoverer and MS Annika were used for database search and data validation.

acquisition and interpretation. These instruments leverage Orbitrap technology, renowned for its high resolving power, mass accuracy, and dynamic range, all of which are crucial for CLMS, where crosslinked peptides are often present in low abundance and require precise identification.

The Astral and Eclipse Orbitrap instruments present distinct advantages over previous generations of mass spectrometers in terms of sensitivity, speed, and operational features, which impact their respective performance in CLMS workflows. The Orbitrap Eclipse is equipped with an advanced ion-routing multipole and a versatile scan strategy that allows for rapid switching between different fragmentation techniques, such as higher-energy collisional dissociation (HCD) and collision-induced dissociation (CID). Furthermore, it offers an advanced ion management technology (AIM+), which enhances mass selection precision, contributing to improved detection of low-abundance peptides, and it features an advanced peak determination algorithm, which improves precursor annotation in data-dependent experiments, further enhancing the detection of low-abundance crosslinked peptides. This flexibility supports broader sequence coverage, providing better identification of peptides[25] and therefore potentially also of crosslinked species. On the other hand, the Astral instrument utilizes a novel multi-reflection time-of-flight (MR ToF) analyzer with isochronous drift in elongated ion mirrors[21,22]. This innovative design significantly improves resolving power while maintaining high sensitivity, which is especially important for low-abundance peptide species like crosslinks. Furthermore, the Astral employs an "Asymmetric Track Lossless" mode for ion transmission, resulting in nearly lossless ion movement and enhanced sensitivity for data acquisition, making it potentially highly effective for CLMS applications[21,22]. Compared to the Eclipse, the Astral shows notable improvements in sensitivity and throughput, quantifying significantly more peptides per unit time and offering high-quality quantitative measurements across a wide dynamic range[25]. These features would allow for deeper exploration of protein interaction networks, especially in workflows involving challenging, low-abundance crosslinked peptides.

Here, we compare the performance of the Orbitrap Eclipse and Astral mass spectrometers in the context of CLMS workflows. We show that the Astral outperforms the Eclipse, achieving up to 40% more unique crosslink identifications through higher MS1 sensitivity, faster scan speeds, and efficient detection of low-abundance precursors. Single HCD consistently enhances performance on the Astral, while gradient and column optimizations further improve crosslink detection in enriched or purified samples. Together, these findings highlight how instrument choice, fragmentation strategy, and chromatographic design directly impact CLMS outcomes. By highlighting the respective strengths of each instrument, we aim to provide researchers with a clearer understanding of the optimal choice for their specific CLMS applications.

## Results

To compare the performance of both instruments, we used the same liquid chromatography (LC) setup equipped with a 25 cm IonOpticks Aurora Ultimate column. Cas9-Helo protein was crosslinked with either PhoX (DSPP) or DSSO and used as a quality control (QC) sample throughout the entire experimental series. To ensure that the difference in results is unique to the instruments, the QC samples were produced in a bigger batch of 100 µg total protein amount for each crosslinker and frozen in aliquots for long-term storage. Aliquots from the same batch were always injected on both instruments equally to reduce variability coming from the crosslink reaction and sample preparation procedure. Both instruments were equipped with high-field asymmetric-waveform ion-mobility spectrometry (FAIMS) devices for ion filtering and noise reduction during data acquisition. LC setup, gradient design, and acquisition methods were kept as similar as

isotopically labeled reagents, improve specificity and sensitivity, enabling studies of dynamic interactions in native environments[13,14]. CLMS extends to in vivo systems, revealing transient interactions in detail[15–17]. Integrative approaches combining CLMS with computational tools like AlphaFold and AlphaLink with cryo-EM enable comprehensive mapping of interaction networks and conformational dynamics[18–20].

Despite its promise, the success of CLMS heavily relies on the performance of the mass spectrometric instrumentation, which must be capable of managing complex peptide mixtures while delivering high-resolution and accurate mass measurements. Recent advancements in Orbitrap mass spectrometers, particularly the Thermo Scientific Orbitrap Astral[21,22] and Eclipse[23,24] instruments, have introduced new capabilities that can significantly enhance crosslinking data

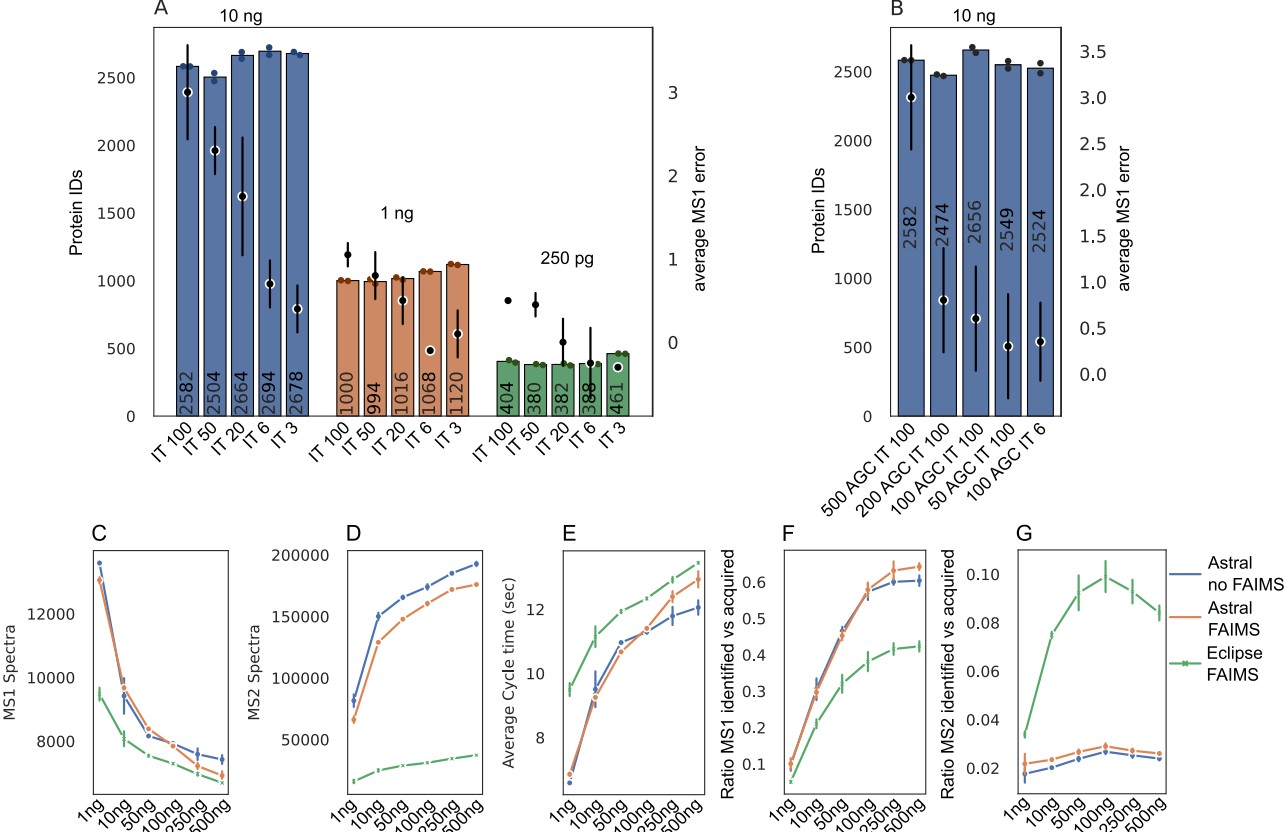

**Fig. 2 | Acquisition method optimization for Orbitrap Astral measurements.**
**A** MS1 injection time optimization using FAIMS and HeLa lysate QC samples at three injection amounts (10 ng, blue; 1 ng, orange; 250 pg, green), with MS1 injection times ranging from 100 ms to 3 ms. **B** Evaluation of automatic gain control (AGC) settings using 10 ng HeLa lysate. **C** Number of MS1 spectra acquired over a 70 min active gradient on the Orbitrap Astral with (orange) and without FAIMS (blue), compared to the Orbitrap Eclipse with FAIMS (green), across a dilution series (1–500 ng of PhoX crosslinked Cas9). **D** Number of MS2 spectra acquired under the same conditions. **E** Average cycle time across all injection amounts. **F** Ratio of identified vs acquired MS1 spectra. **G** Ratio of identified vs acquired MS2 spectra. For all bar plots, the mean number of proteins is shown at the center of each bar. Individual technical replicates ($n = 2$, **C**–**G** $n = 3$) are indicated by black dots. Error bars represent standard deviation, calculated as the average distance from individual data points to the sample mean.

possible for both instruments (Fig. 1). Injection times (IT) and automatic gain control (AGC) and compensation voltages (CV) were optimized for the Orbitrap Astral as shown in Figs. 2, 3 and Supplementary Fig. 2. According to previous publications, CV values for the Eclipse measurements were selected as CV −50 V, −60 V, and −70 V[26].

### Method optimization on the Orbitrap Astral
For initial acquisition method optimization, we evaluated MS1 AGC targets and injection times (IT) using a HeLa lysate QC sample across decreasing injection amounts (Fig. 2A). A series of measurements was performed with a fixed AGC target of 500, varying the MS1 injection time from 100 ms down to 3 ms, using 10 ng, 1 ng, and 250 pg of HeLa digest. While the number of protein identifications remained largely unaffected by changes in injection time, the average MS1 mass error improved substantially, from +3 ppm at 100 ms to +0.5 ppm at 3 ms, consistently across all sample amounts. In a separate experiment using 10 ng of HeLa lysate and a fixed injection time of 100 ms, the AGC target was varied from 500 to 50 (Fig. 2B). Again, the number of protein identifications remained unchanged, but a marked reduction in average MS1 mass error was observed at lower AGC settings. Based on these observations, an AGC target of 500 and a reduced injection time of 6 ms were selected as the optimal settings to balance sensitivity and mass accuracy for subsequent crosslinked sample analyses. Using these optimized acquisition parameters, a dilution series of PhoX-crosslinked Cas9 (1 ng to 500 ng) was analyzed to assess MS1 mass accuracy for crosslinked precursors. Across all injection amounts, MS1

errors remained low and centered near 0 ppm, with increasing injection amounts leading to broader mass error distributions. The widest spread was observed at 500 ng. When the same acquisition series was repeated on the Orbitrap Eclipse, mass errors were consistently offset from zero, with reduced distributional spread compared to the Astral, particularly at higher injection amounts (Supplementary Fig. 11A).

To further enhance crosslink identification, FAIMS compensation voltage (CV) settings were optimized. Single CV values ranging from −30 V to −90 V were first evaluated individually, revealing −48 V as the best-performing setting, yielding up to 326 unique residue pairs (Fig. 3C). Charge-state-dependent feature detection showed that charge states above +3 were maximally detected at CVs between −50 V and −48 V, while charge states of +1 and +2 reached their maxima around −30 (Fig. 3D). Next, all pairwise and triplet combinations of CV values were analyzed using the Upset plot function in Python to determine optimal combinations based on crosslink yield and redundancy (Fig. 3G–I). Three CV triplets were prioritized: −48 V/−60 V/−75 V (highest number of crosslinks with minimal overlap), −48 V/−55 V/−90 V (minimal total overlap), and −40 V/−48 V/−60 V (maximum overlap). These were benchmarked against a standard QC combination of −48 V/−60 V/−80 V using 100 ng of crosslinked material. The −48 V/−60 V/−75 V combination yielded the highest number of unique residue pairs (1272), outperforming the high-overlap combination by 20% (Supplementary Fig. 2). Notably, optimal CV values may vary across FAIMS devices, necessitating individual calibration per instrument. The optimal CV combination

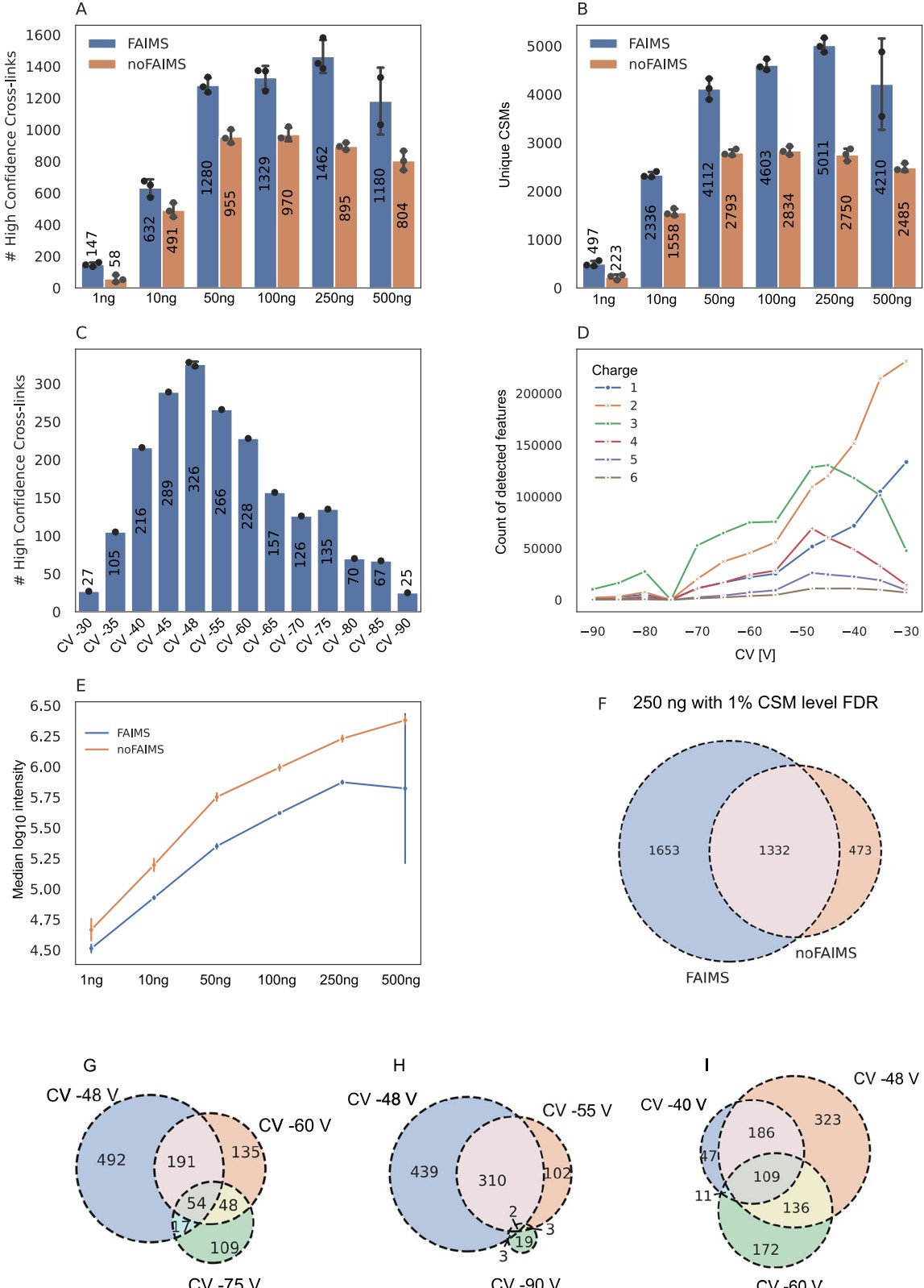

(−48 V/−60 V/−75 V) was used to assess the impact of FAIMS on crosslink identifications relative to non-FAIMS acquisitions on the Orbitrap Astral. At the unique residue pair level (Fig. 3A) and cross-link spectrum match (CSM) level (Fig. 3B), FAIMS enhanced identifications by an average of 30%, with a maximum at 250 ng injection. In contrast, non-FAIMS measurements peaked between 50-100 ng. These trends were mirrored at the CSM level. The underlying

performance improvement was examined by quantifying MS1 apex intensities of crosslinked precursors and comparing their distributions across FAIMS and non-FAIMS acquisitions (1–500 ng). FAIMS consistently enabled detection of lower-abundance precursors due to reduced background "noise" and complexity of the ion distribution, with a more pronounced benefit at higher injection amounts (Fig. 3E; full distributions in Supplementary Fig. 9B). For 250 ng

**Fig. 3 | Impact of FAIMS on crosslinking performance for PhoX-crosslinked Cas9 analyzed on the Orbitrap Astral using a active gradient of 70 min.**
**A** Number of unique crosslinks identified across a dilution series (1–500 ng injection) with (blue) and without FAIMS (orange). **B** Number of crosslink spectrum matches (CSMs) at 1% CSM level FDR across the same dilution series. **C** Optimization of compensation voltage (CV) settings on the Astral for PhoX-crosslinked Cas9 ranking from CV −30 to −90 V. **D** Charge state distribution of detected features across CV settings with +1 (blue), +2 (orange), +3 (green), +4 (red), +5 (violet) and +6 (braun). **E** Median $\log_{10}$ MS1 precursor intensities of identified features for FAIMS (blue) and no FAIMS (orange) measurements across the injection series. Full intensity distributions can be inspected in Supplementary Fig. 9B. **F** Overlap of unique crosslinks identified with (blue) and

without FAIMS (orange) at 1% CSM-level FDR. G-I. Based on an upset plot of all CV measurements, the three most promising CV combinations were selected for further evaluation. Combination 1 **G** CV −48 V (blue), −60 V (orange), −75 V (green) represent the number of crosslinks with the least overlap, combination 2 **H** CV −48 V (blue), −55 V (orange), −90 V (green) the least overall overlap and combination 3 **I** CV −40 V (blue), −48 V (orange), −60 V (green) the highest overlap. The unique crosslinks plotted in venn diagrams have been summed up from CSMs with 1% CSM level FDR. For all bar plots, the mean number of residue pairs/CSMs is shown at the center of each bar. Individual technical replicates ($n = 3$) are indicated as black dots. Error bars represent standard deviation, calculated as the average distance from individual data points to the sample mean.

injections, FAIMS-specific identifications constituted 48% of the total, with 39% overlap and only 14% unique to non-FAIMS (Fig. 3F).

Method development revealed substantial performance differences between FAIMS and non-FAIMS configurations, as well as between the Orbitrap Astral and Orbitrap Eclipse platforms. At low sample loads, the Astral acquired significantly more MS1 spectra than the Eclipse; however, this difference diminished at higher injection amounts (Fig. 2C). In contrast, MS2 spectral acquisition exhibited the opposite trend: while both instruments produced similar numbers of MS2 scans at low loads, the Astral outperformed the Eclipse markedly at higher loads, acquiring more MS2 spectra under all tested conditions (Fig. 2D). This increased acquisition rate is further supported by shorter cycle times on the Astral, which enables faster data collection compared to the Eclipse (Fig. 2E). Comparison of the Astral with and without FAIMS revealed minimal differences in MS1 scan counts and cycle times across all injection amounts. However, FAIMS resulted in a slight reduction in MS2 scans compared to non-FAIMS measurements. When evaluating the proportion of identified MS1 spectra relative to total acquired scans, the Astral consistently outperformed the Eclipse across all injection amounts, indicating superior precursor feature detection and improved MS1 spectral utility for crosslink identification (Fig. 2F). Notably, this identification efficiency increased with sample load, underscoring the instrument's sensitivity and data quality at higher concentrations. At MS2 level, the ratio of identified to acquired spectra exhibited an inverse pattern. The Astral generated more MS2 spectra than it identified across all injection amounts, reflecting its rapid acquisition capability (Fig. 2G). Conversely, the Eclipse displayed higher identification efficiency per MS2 scan, with this ratio increasing with injection amount and peaking at 100 ng. FAIMS had minimal impact on MS1 and MS2 identification ratios on the Astral, suggesting that the observed performance gains are not due to changes in acquisition rates. Rather, the improvements are attributed to the enhanced signal-to-noise ratio and charge-state filtering enabled by FAIMS, which preferentially transmits higher-charged precursors ($z > +3$) in our crosslink acquisition method. This is reflected in the observed shift toward low-abundance crosslinked peptides in FAIMS datasets (Fig. 3E), highlighting the combined benefits of FAIMS and Astral's high sensitivity for CLMS. This benefit was shown in the literature before on other Orbitrap instruments for example on the Exploris 480 or Fusion Lumos with FAIMS enhancing the crosslink identifications for low injections amounts (10–100 ng) compared to non-FAIMS measurements[26–28].

### Instrument comparison

The comparison of the Astral and Eclipse instruments for the identification of crosslinks using both non-cleavable (PhoX) and cleavable (DSSO) crosslinkers are illustrated by injecting dilution series from 1 ng to 500 ng of crosslinked Cas9 protein for both crosslinkers on both instruments in parallel. The maximum number of unique crosslinks was achieved at 250 ng for PhoX and 500 ng for DSSO (Fig. 4A, C). In both cases, the Astral mass analyzer significantly outperformed the Eclipse by over 40%. Interestingly, 192 and 121 unique residue pairs

(URPs) could be identified with 1 ng injection amount for PhoX and DSSO, respectively. The overlap of unique crosslinks identified by Astral and Eclipse at 1% CSM-level FDR for the PhoX crosslinker, based on concatenated CSMs is 40% of crosslinks (970 links), with 47% unique to Astral and 13% unique to Eclipse (Fig. 4B) and for DSSO data 45% overlap (526 links), with 49% unique to Astral and only 5% unique to Eclipse (Fig. 4D). This improvement might be attributed to the enhanced sensitivity of Astral's MR ToF analyzer, which is particularly effective in maintaining resolution for low-abundance species. The observed 40% increase in crosslink identifications on the Orbitrap Astral compared to the Orbitrap Eclipse cannot be solely attributed to acquisition speed. While the Astral indeed acquires substantially more MS2 spectra due to its faster scan rate (Fig. 2E), its identification success rate per acquired MS2 scan is actually lower than that of the Eclipse. A notable advantage of the Astral, however, lies in its higher MS1-level identification success rate, as evidenced in Fig. 2G. This suggests that the Astral's capacity to produce higher-quality precursor information may contribute to its superior overall performance. Nonetheless, we propose that the primary advantage of the Astral lies in its enhanced sensitivity (Supplementary Fig. 9A). This increased sensitivity enables the detection of lower-abundance crosslinked peptides, which translates directly into improved identification rates. Sensitivity appears to be a key parameter for CLMS performance, a conclusion further supported by the ~30% identification gain observed when implementing FAIMS on the same instrument. The similarity of score distributions between the Astral and Eclipse datasets (Supplementary Fig. 11B) indicates that improved spectral quality alone does not account for the higher identification yield, reinforcing the role of sensitivity in driving the Astral's performance advantage. To assess the biological relevance of the additional low-abundance crosslinks identified by the Astral, we evaluated crosslinked peptide identifications stratified by precursor abundance. Peptide features were matched to CSMs using compensation voltage, charge state, and m/z, with a mass tolerance of 0.01 and retention time alignment to improve matching accuracy. Apex intensities were classified into three categories: low ($<1 \times 10^5$), medium ($1 \times 10^5$ to $1 \times 10^7$), and high ($>1 \times 10^7$). For each category and across all injection amounts, the identified crosslinks were mapped onto a predicted AlphaFold3 structure of Cas9. Euclidean distances between linked residues were calculated and displayed as histograms to evaluate abundance-dependent trends in crosslink lengths, while xiView[29] circle plots were used to visualize changes in spatial coverage on the protein (Supplementary Fig. 8A–G). The proportion of short-distance crosslinks (<20 Å, the theoretical upper limit for PhoX) across abundance categories is shown in Fig. 4E. Low-abundance crosslinks showed an increasing proportion of distances below 20 Å with higher injection amounts, suggesting improved coverage of structurally relevant interactions at increased sample load. In contrast, medium- and high-abundance crosslinks displayed the highest proportion of short-distance interactions at the lowest injection amount (1 ng), with a progressive increase in overlength crosslinks at higher loads. Structural mapping revealed that at low injection amounts, high-abundance crosslinks predominantly localize to protein

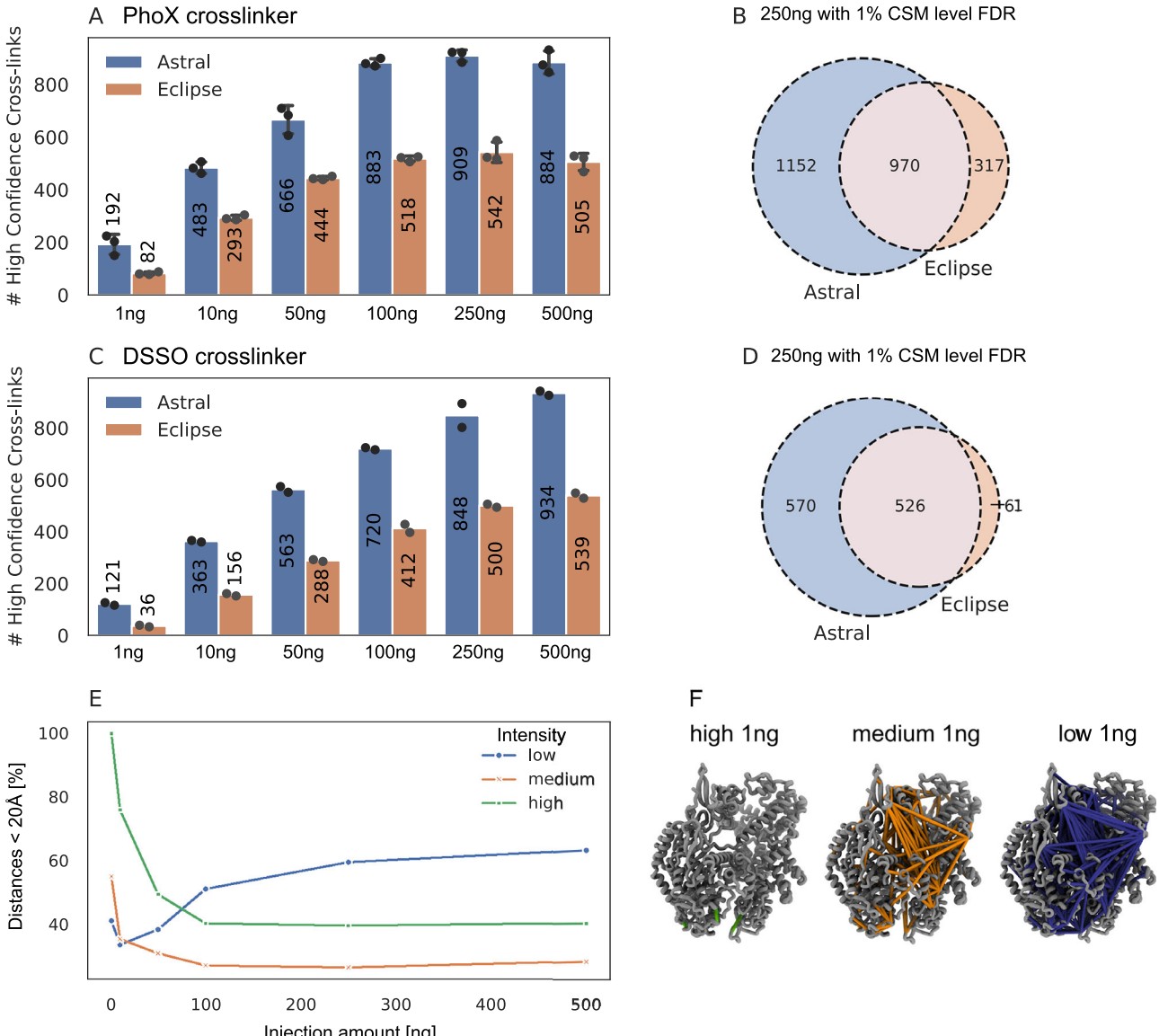

**Fig. 4 | Comparison of Astral and Eclipse crosslinking data for the non-cleavable crosslinker PhoX and the cleavable crosslinker DSSO using FAIMS.**
**A** Dilution series of Cas9 crosslinked with PhoX, ranging from 1 ng to 500 ng, analyzed on both the Eclipse (orange) and Astral (blue) mass spectrometers in triplicate. **B** Overlap of unique crosslinks identified by Astral and Eclipse at 1% CSM-level FDR for the PhoX crosslinker with 250 ng injection amount, based on concatenated CSMs. **C** Dilution series of Cas9 crosslinked with DSSO, ranging from 1 ng to 500 ng, analyzed in duplicate. **D** Overlap of unique crosslinks identified by Astral and Eclipse at 1% CSM-level FDR for DSSO with 250 ng injection amount. **E** Proportion of crosslinks within 20 Å, based on an AlphaFold3-predicted

Cas9 structure, ranked by MS1 apex intensity (low: $<1 \times 10^5$, medium: $1 \times 10^5 - 1 \times 10^7$, high: $>1 \times 10^7$) and injection amount (Astral). Low-abundance crosslinks (blue) are enriched in short-distance links, while medium (orange) and high (green) abundance crosslinks increasingly include over-length interactions at higher injection amounts. **F** Structural mapping of crosslinks identified at 1 ng injection, categorized by abundance: low (blue), medium (orange), and high (green), projected onto the predicted Cas9 structure. In all bar plots, the mean number of residue pairs is shown at the center of each bar. Individual technical replicates are depicted as black dots (PhoX: $n = 3$; DSSO: $n = 2$). Error bars represent standard deviation, calculated as the average distance from individual data points to the sample mean.

surface regions, whereas medium- and low-abundance crosslinks increasingly span the protein core (Fig. 4F).

These findings suggest that crosslink abundance is closely related to crosslinking efficiency and structural accessibility. Surface-exposed residues, being more accessible and reactive, tend to yield high-abundance crosslinks. In contrast, crosslinks buried within the protein or representing transient protein-protein interactions occur less frequently and are thus detected at lower abundance. Importantly, these low-abundance crosslinks likely reflect structurally or functionally significant interactions. This insight opens new methodological avenues: by leveraging MS1 precursor intensity as a proxy for structural relevance, acquisition strategies could be refined to preferentially target low-abundance crosslinked species. Such an approach, e.g., MS1

intensity-based internal fractionation, could enhance the detection of structurally informative or transient crosslinks, improving both sensitivity and interpretability in CLMS experiments.

**Fragmentation strategy comparison on Astral and Eclipse**
To evaluate the impact of fragmentation strategy on crosslink identification efficiency, we compared single HCD and stepped HCD on the Orbitrap Astral across a dilution series (1–500 ng) of Cas9 samples crosslinked with either PhoX or DSSO. Previous reports have suggested a benefit of stepped HCD for cleavable crosslinkers[30]; however, our results indicate that, on the Astral, single HCD consistently outperforms stepped HCD for both crosslinkers (Fig. 5). For PhoX-crosslinked samples, the number of URPs was highest at 250 ng

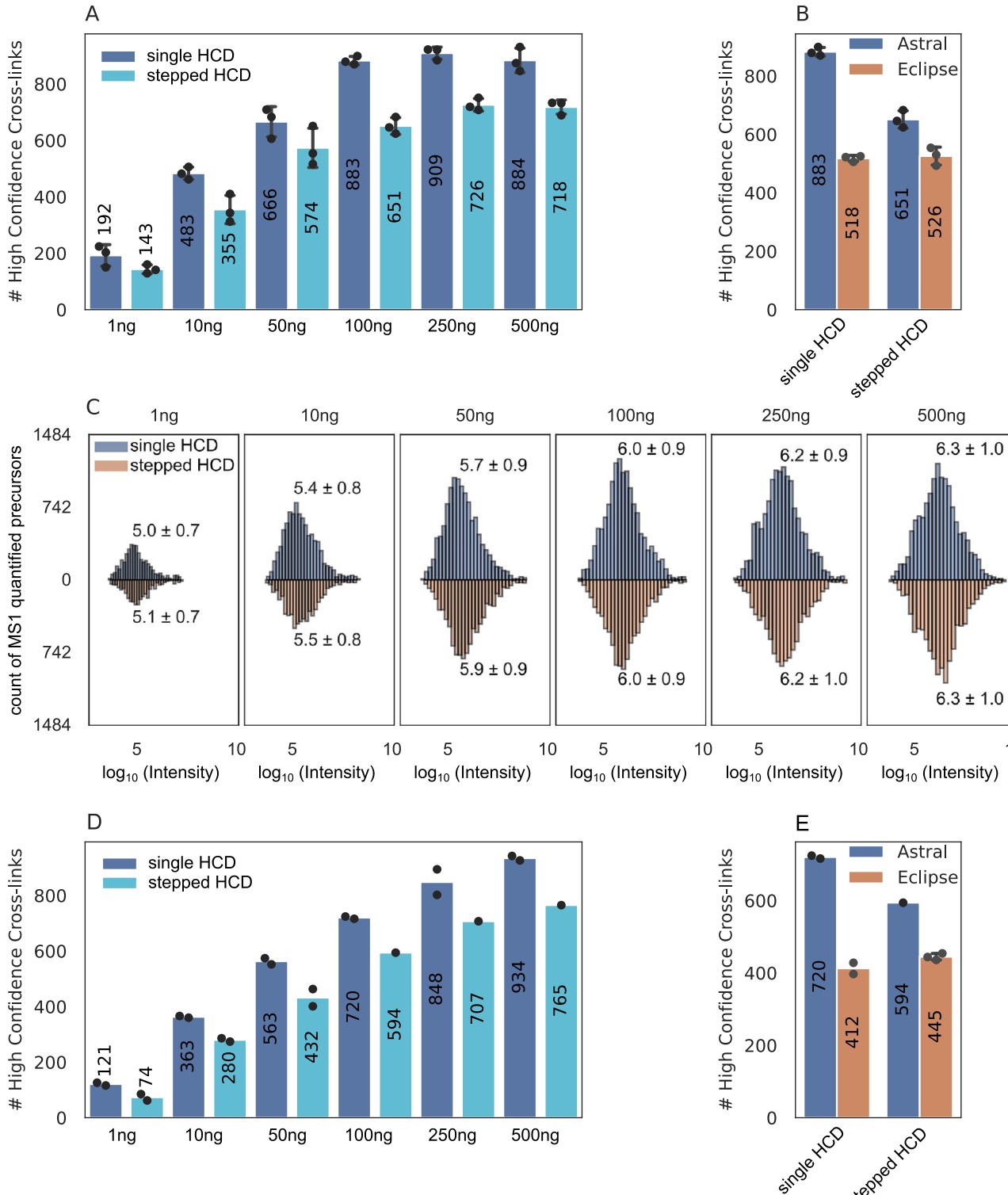

**Fig. 5 | Comparison of single versus stepped HCD fragmentation for PhoX (non-cleavable) and DSSO (cleavable) crosslinkers on Orbitrap Astral with FAIMS.** **A** Comparison of unique crosslinks identified using single HCD (dark blue, HCD 32%) and stepped HCD (light blue, HCD 25,27,32%) across a dilution series of PhoX-crosslinked Cas9 on the Orbitrap Astral. **B** Differential performance of single and stepped HCD at 100 ng injection analyzed on both Astral and Eclipse instruments. **C** MS1 precursor abundance distributions for the PhoX dilution series, comparing single HCD (blue) and stepped HCD (orange). Medians and standard deviations (calculated as the average distance from individual values to the sample mean) are displayed above each distribution. **D** Comparison of unique crosslinks identified using single (dark blue, HCD 30%) and stepped HCD (light blue, HCD 21,25,32%) across a dilution series of DSSO-crosslinked Cas9 on the Orbitrap Astral. **E** Comparison of single and stepped HCD at 100 ng DSSO-crosslinked Cas9 on Astral and Eclipse. In all bar plots, the mean number of residue pairs is shown at the center of each bar. Individual technical replicates are depicted as black dots (PhoX: $n = 3$; DSSO: $n = 2$/ $n = 1$). Error bars represent standard deviation, calculated as the average distance from individual data points to the sample mean.

injection with single HCD yielding 909 URPs versus 726 URPs with stepped HCD, corresponding to a 25% increase. This benefit was consistent across the dilution series, with identification gains ranging from 19 to 26% in favor of single HCD (Fig. 5A). Similar trends were observed with DSSO-crosslinked samples, where single HCD outperformed stepped HCD by up to 39% at the lowest injection amount (1 ng), and by 18% at the highest (500 ng). At 250 ng, single HCD yielded 848 URPs compared to 707 URPs with stepped HCD (Fig. 5D). Importantly, this performance advantage was specific to the Astral. On the Orbitrap Eclipse, the difference between fragmentation methods was negligible or slightly favored stepped HCD. For instance, PhoX samples at 100 ng showed a non-significant 1.5% increase in identifications with stepped HCD, while DSSO samples showed a 7.4% gain (Fig. 5B, E). This advantage may stem from the Astral's highly efficient ion transmission and fast scan capabilities. Single HCD allows acquisition speeds of up to 220 Hz with 2 ms MS2 injection time (Thermo Fisher Scientific Poster, Jimmy M. Garland et al. 2025, https://lcms.cz/labrulez-bucket-strapi-h3hsga3/po_335_state_art_ms_astral_zoom_po335_asms2025_na_en_e61daa165c.pdf), faster than the ~80 Hz observed for stepped HCD, which inherently includes multiple collision energies per scan. This increased scan rate likely enhances sampling density and improves duty cycle utilization, particularly critical for low-abundance precursors. In our experiments the MS2 scan rate reached ~35 Hz with 20 ms and 250 ng injection amount for single HCD and 22 Hz for stepped HCD (Supplementary Fig. 11C).

To investigate whether differences in spectral quality contributed to the performance discrepancy, representative crosslinked peptides identified by both methods and both instruments were compared (Supplementary Fig. 10A–D). Although stepped HCD produced more high m/z fragments and slightly more complex spectra, overall sequence coverage and fragment quality were comparable between the two methods. Score distributions between these two methods on Astral were similar as well and showed only a minimal advantage for single HCD (Supplementary Fig. 11D). These results suggest that the gain in identifications with single HCD does not stem from improved fragmentation per se. Additionally, log-transformed intensity distributions were analyzed (Fig. 5C). For lower injection amounts, stepped HCD spectra showed slightly higher median intensities. Still, this effect diminished at higher sample loads, indicating that intensity differences do not explain the consistent performance benefit of single HCD. Instead, the critical factor appears to be scan speed (Supplementary Fig. 11C). The Astral's advantage arises from its ability to capitalize on the simplicity and efficiency of single HCD fragmentation, whereas the Eclipse, equipped with advanced ion optics and ion-routing technologies, maintains high identification rates under both fragmentation strategies. The Eclipse's precise ion control likely mitigates the drawbacks of the stepped approach, explaining its smaller performance differential. These findings underscore the need for instrument-specific optimization of fragmentation strategies in CLMS. On fast-scanning platforms like the Orbitrap Astral, maximizing duty cycle through single HCD enhances identification performance, particularly for low-abundance analytes. Conversely, on systems like the Orbitrap Eclipse, the flexibility and stability of ion handling enable effective use of stepped fragmentation without significant compromise in sensitivity or throughput.

## Gradient optimization for enhanced crosslink identification

In shotgun proteomics, longer chromatographic separation times have been shown to improve protein identifications for injection amounts up to 1 µg. To investigate whether this observation holds true for crosslinking experiments, a sample using 250 ng of Cas9-Helo crosslinked with PhoX was analyzed across active gradient lengths ranging from 10 to 180 min. The results demonstrated a dramatic increase in the number of URPs from 45 URP with a 10-min gradient to 436 URP with a 180-min gradient, equating to an 868.9% increase (Fig. 6A). CSM

counts similarly rose from 110 to 5055 CSMs across the same gradient range, corresponding to a 4495.5% increase (Fig. 6B). This substantial improvement can be explained by the growing number of supporting CSMs per unique crosslink with increasing gradient time: while short gradients yielded only 2.5 CSMs per URP, the longest gradient delivered up to 12 CSMs per URP (Fig. 6D). Notably, the number of crosslinks identified per minute peaked early during the shortest gradients (10–20 min) and declined steadily thereafter (Fig. 6C), indicating that while longer gradients maximize total identifications, the efficiency of identification per unit of time diminishes beyond this early peak. Although extending gradients beyond 180 min may yield further gains in crosslink identifications, this would come at the cost of longer acquisition times and increased financial and instrumental resource usage. These findings highlight the trade-off between maximizing identification numbers and maintaining experimental throughput.

To evaluate whether these observations apply to more complex sample backgrounds, the gradient optimization was repeated using PhoX-crosslinked *E. coli* ribosomes in a HeLa lysate background at a 1:2 ratio. Again, gradient lengths from 10 to 180 min were tested. For this complex sample, monolink identifications increased rapidly between 10 and 70 min and plateaued afterwards (Fig. 6E). However, unique crosslink identifications remained low overall, with a maximum of 8 URPs at 180 min, likely due to the sample complexity and the absence of any crosslink enrichment strategy. The same pattern was observed for general protein identification using *E. coli* ribosome proteins within the complex matrix, where protein identifications rose steeply from 70 to 152 between 10 and 70 min and then levelled off (Fig. 6F). These results indicate that while longer gradients can increase identification depth in enriched or purified samples, their utility in complex backgrounds without prior fractionation or enrichment is limited. For such complex samples, a 70-min gradient represents a balanced compromise between identification yield and instrument run time. In summary, longer gradients significantly enhance peptide and crosslink identifications for clean or enriched samples, driven by increased CSM support per crosslink. However, in complex samples such as lysates, this benefit is quickly capped due to the limited dynamic range and complexity of the sample, suggesting that gradient extension alone is insufficient. Instead, improving identifications in such contexts may require enrichment strategies or purification of the crosslinked protein complexes, rather than simply increasing gradient duration.

## Column comparison

We briefly tested a 50 cm PepMap column (Thermo Fisher Scientific) and a 25 cm Aurora Ultimate (IonOpticks) analytical reverse-phase column to evaluate the optimal separation performance for crosslinked peptides. Both columns use C18 as a stationary phase and are suitable for nano-flow setups but differ substantially in particle size, pore size, length, and pressure limit. The key parameters of both columns are summarized in Supplementary Table 1. The PepMap and Aurora columns exhibit distinct characteristics that make them suitable for different analytical applications. The PepMap column, with its longer length of 50 cm and a particle size of 2 µm, is ideal for achieving high-resolution separations in complex samples, particularly when comprehensive profiling is required. Its pore size of 100 Å makes it well-suited for smaller analytes, although its pressure limit of 1500 bar may necessitate careful monitoring during high-flow operations. Conversely, the Aurora column, with its shorter length of 25 cm and smaller particle size of 1.7 µm, delivers rapid and efficient separations, making it advantageous for high-throughput analyses. Its larger pore size of 120 Å supports the analysis of larger biomolecules such as crosslinked peptides. Furthermore, the Aurora column's higher pressure tolerance (>1700 bar) offers greater robustness for demanding workflows. These potential advantages for crosslinked peptides resulted in 779 URPs for 100 ng injection of crosslinked Cas9 with PhoX, compared to 560 URPs acquired using the PepMap column. The

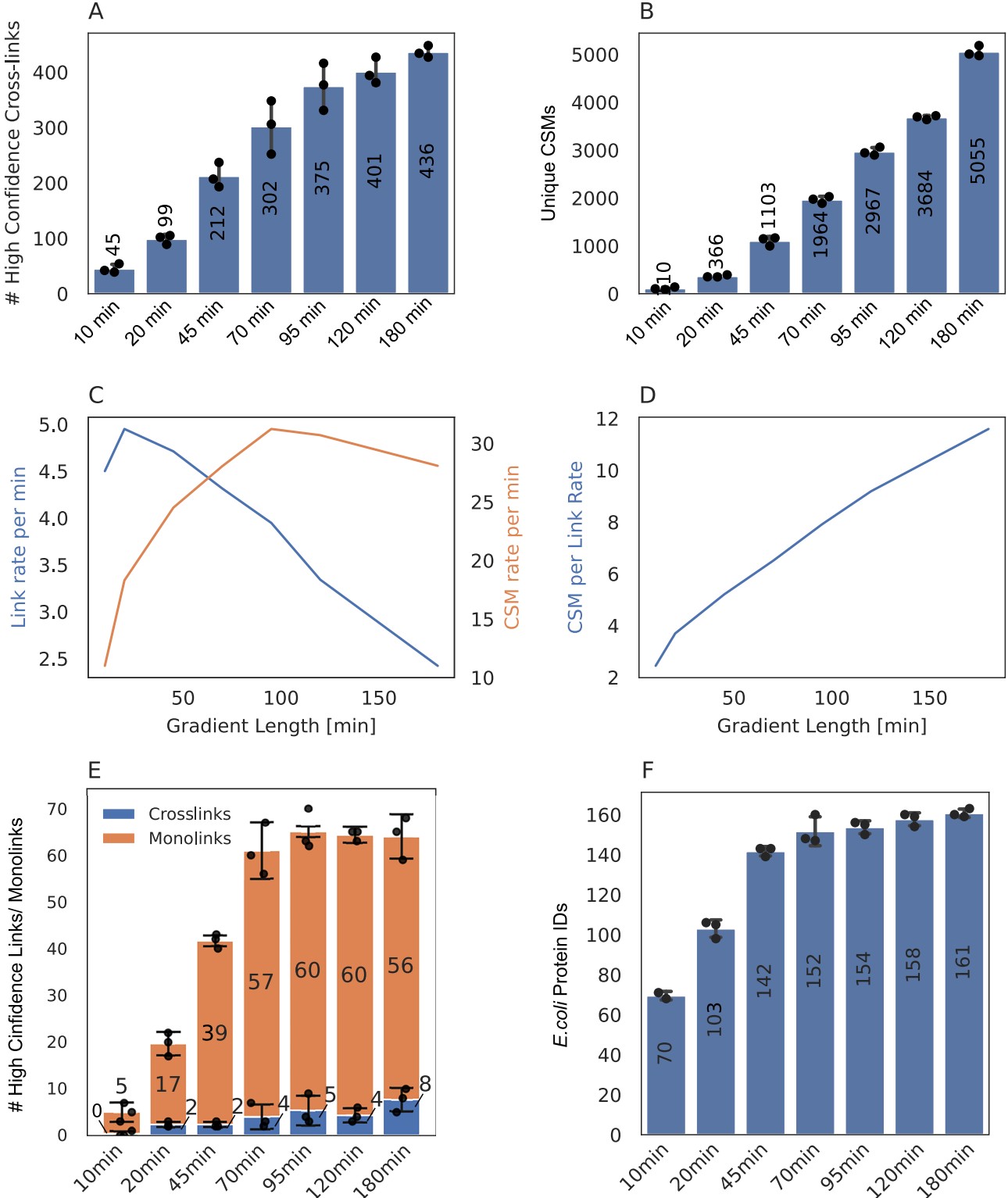

**Fig. 6 | Gradient length optimization for PhoX-crosslinked Cas9 (250 ng) and crosslinked *E. coli* ribosomes in a HeLa lysate background (1:2), analyzed on the Orbitrap Astral with FAIMS. A** Number of unique Cas9 crosslinks identified across gradient lengths ranging from 10 to 180 min active gradient time. **B** Unique crosslink spectrum matches (CSMs, 1% CSM level FDR) in dependence of active gradient time. **C** Line plot showing the number of unique crosslinks per minute (blue) and CSMs per minute (orange) as a function of gradient length. **D** Average number of CSMs per unique crosslink as a function of gradient time. **E** Comparison of unique crosslinks and monolinks identified as a function of gradient length from PhoX crosslinked *E. coli* ribosomes in a HeLa lysate background (1:2 mix). The injection amount was constant across conditions at 250 ng. **F** Number of protein identifications from linear (non-crosslinked) *E. coli* ribosome searches. For all bar plots, the mean value of residue pairs/CSMs/monolinks/protein IDs is shown at the center of each bar. Individual technical replicates (*n* = 3) are represented by black dots. Error bars denote standard deviation, calculated as the average distance from each data point to the sample mean.

Aurora column therefore outperformed PepMap by 28% (Supplementary Fig. 3). Quantitation in Skyline of 5 selected crosslinked peptides across a 70 min gradient revealed smaller full-width half maximum values for the aurora column, resulting in sharper peaks and therefore higher intensities of the selected peptides (Supplementary Fig. 4A, B). Higher intensities seem to be the key point in performance difference between both columns and lead to an overall better performance of the aurora column. Furthermore, better peptide separation could be observed for the aurora column as seen in Supplementary Fig. 5. The chromatographic resolution of the aurora column outperformed the PepMap column despite the longer separation path of 50 cm vs. 25 cm. Effective separation requires peptides to enter the pores of the column's porous particles. When their hydrodynamic diameter is too large, access to the internal volume is reduced, causing diminished retention, peak broadening, or tailing[31]. To utilize at least 50% of the pore volume, the pore size should be three to five times larger than the peptides/proteins hydrodynamic diameter[32,33]. 90% of theoretical human tryptic peptides have a molecular weight of less than 3 kDa (with two missed cleavages, only 10% exceed 5.6 kDa) and hydrodynamic diameters below 100 Å[31]. Crosslinked peptides typically have larger molecular weights because they are formed by the combination of two peptides along with the added mass of the crosslinker[34]. As a result, they require larger pore sizes for effective separation compared to linear peptides. The larger pore size of the Aurora column (120 Å vs 100 Å PepMap) likely enhanced here the binding of larger crosslinked peptides. Despite the pore size, the Aurora column most likely also benefits from the smaller particle size of 1.7 μm instead of 2 μm, because column efficiency is usually proportional to the particle diameter[31,35], hence smaller particle size improves the separation of peptides. Furthermore, the IonOptics column features enhanced ionization efficiency due to its smaller emitter diameter (~6 μm) compared to the PepMap emitter setup using a fused silica emitter (10 μm) with integrated liquid junction (Bruker).

## Discussion

Our results demonstrate that the Orbitrap Astral significantly improves CLMS performance compared to the Orbitrap Eclipse, yielding up to 40% more unique crosslink identifications. This increase is primarily driven by higher MS1-level identification efficiency and improved sensitivity on the Astral, particularly for low-abundance precursors, rather than differences in spectral quality or MS2 identification success. The Astral's ability to detect lower-intensity crosslinked peptides enabled deeper structural coverage, as low-abundance crosslinks were enriched for short-distance interactions and often mapped to internal or less accessible protein regions. In contrast, high-abundance crosslinks localized predominantly to surface-exposed areas, suggesting that crosslink abundance reflects structural accessibility and crosslinking efficiency. These findings highlight the biological relevance of low-abundance crosslinks and suggest that MS1 intensity could guide targeted acquisition strategies to enhance structural insight.

Contrary to previous reports favoring stepped HCD for cleavable crosslinkers, single HCD fragmentation consistently outperformed stepped HCD on the Astral across all injection amounts for both DSSO and PhoX. This effect was most pronounced at low sample inputs and likely stems from the substantially higher scan speed of single HCD. On the Eclipse, fragmentation strategy had minimal impact, likely due to its advanced ion control systems that mitigate the complexity of stepped fragmentation. This emphasizes the need for instrument-specific optimization in CLMS workflows. Additionally, gradient optimization experiments further revealed that longer separation times substantially increase crosslink identifications in simple or enriched samples, largely due to increased CSM support per link. However, this benefit plateaus beyond 70 min in complex lysates, where enrichment or complex isolation becomes a more effective strategy than increasing gradient length alone.

**Table 1 | Special reagents used for comparing the Orbitrap Astral and Orbitrap Eclipse mass spectrometers**

| Reagent name | Catalogue number | Supplier |
|---|---|---|
| Cas9 from *S. pyogenes* fused with a Halo-tag | In house | Deng et al.[60] |
| Trypsin Gold | V5280 | Promega |
| PhoX (DSPP) | A52286 | Thermo Fisher Scientific |
| DSSO | A33545 | Thermo Fisher Scientific |
| Aurora Ultimate | AUR3-25075C18 | IonOpticks |
| PepMap | DNV75500PN | Thermo Fisher Scientific |
| Fused silica emitter (ID 10 μm, OD 150 μm) | PSFSE 10 (1893527) | Bruker |

Together, these findings demonstrate that sensitivity, scan speed, and targeted method optimization are critical factors in maximizing CLMS performance. The Astral, with its high-speed acquisition and superior low-abundance detection, is particularly well-suited for applications requiring high depth and structural resolution. The data presented in this study emphasize the impact of advanced instrumentation, fragmentation techniques, and optimized gradients on CLMS performance. Researchers can utilize these insights to better choose appropriate instrumentation and optimize workflows for their specific biological questions, ultimately contributing to a more detailed understanding of protein interactions.

## Methods
### Reagents
Table 1.

### Crosslinking reaction for Cas9
Cas9-Halo protein was crosslinked using either PhoX or DSSO. All crosslinkers were prepared as stock solutions at a concentration of 50 mM in dry Dimethyl Sulfoxide (DMSO). For the crosslinking reaction, Cas9 was diluted in 50 mM HEPES to achieve a final protein concentration of 1 μg/uL and a crosslinker was added to a final concentration of 1 mM (PhoX) or 0.2 mM (DSSO). After a 45-min incubation on ice, the reactions were stopped using 100 mM Tris buffer. Each crosslink reaction was prepared in parallel with the same conditions and buffers.

### In-solution digest
For the in-solution digest, proteins were reduced using 10 mM Dithiothreitol (DTT) for 30 min at 50 °C, followed by water bath sonication for 10 min and finally alkylated using 50 mM Iodoacetamide (IAA) for 30 min in the dark. The digest was performed using Trypsin (1:100, enzyme-to-protein ratio). The mixture was incubated overnight at 37 °C to facilitate complete digestion. The digestion process was terminated by the addition of 10% Trifluoroacetic Acid (TFA), adjusting to a final concentration of 0.2%.

### Mass spectrometry
LC-MS/MS analysis was performed using an Orbitrap Eclipse or Orbitrap Astral mass spectrometer with high-field asymmetric ion mobility spectrometry (FAIMS) interface (FAIMS Pro Duo, Thermo Fisher Scientific) coupled with an EASY-Spray source and Vanquish Neo UHPLC system (Thermo Fisher Scientific). A trap column PepMap C18 (5 mm × 300 μm ID, 5 μm particles, 100 Å pore size) (Thermo Fisher Scientific) and an analytical column PepMap C18 (500 mm × 75 μm ID, 2 μm, 100 Å) (Thermo Fisher Scientific) or Aurora Ultimate (250 mm × 75 μm ID, 1.7 μm, 120 Å)(Ion Opticks) were employed for separation. The column temperature was set to 50 °C. Sample loading

was performed using 0.1% trifluoroacetic acid in water with a flow rate of 25 μL/min. Mobile phases used for separation were as follows: A 0.1% formic acid (FA) in water; B 80% acetonitrile, 0.1% FA in water. Peptides were eluted using a flow rate of 230 nL/min (PepMap) or 300 nL/min (Aurora), with the following gradient: from 2 to 37% phase B in 60 min, from 37 to 47% phase B in 10 min, from 47 to 95% phase B in 1 min, followed by a washing step at 95% for 4 min, and re-equilibration of the column. The gradient was altered for the gradient optimization experiments to facilitate longer gradients.

The mass spectrometry settings on the Astral mass spectrometer were set as follows: FAIMS separation was performed with the following settings: inner and outer electrode temperatures were 100 °C, FAIMS carrier gas flow was 3.5 L/min, compensation voltages (CVs) of −48, −60, and −75 V were used in a stepwise mode during the analysis. The FAIMS CV values were measured in a range from −30 to −90 in single measurements during the CV optimization experiment. −48 V has been selected instead of −50 V according to the paper of Bubis et al.[36]. The ion transfer tube temperature was set to 275 °C. The mass spectrometer was operated in a data-dependent mode with cycle time 1s, using the following full scan parameters: m/z range 375–1300, nominal resolution of 180,000, with a target of 500% charges for the automated gain control (AGC), and a maximum injection time of 6 ms. For HCD MS/MS scans, single normalized collision energy (NCE) of 32% (PhoX) and 30% (DSSO) was used for single HCD experiments and stepped HCD values of 25%, 27%, 32% for PhoX[37] and 21%, 25%, 32% for DSSO[30,38]. Precursor ions were isolated in a 1.6 Th (±0.8 Th) window with no offset and accumulated for a maximum of 20 ms or until the AGC target of 500% was reached. Precursors of charge states from 3+ to 6+ were scheduled for fragmentation. Previously targeted precursors were dynamically excluded from fragmentation for 15 s. The sample load was typically in a range of 1 ng to 500 ng as indicated in the respective figure or with 100 ng for the column comparison and 250 ng for the gradient optimization experiments. Detailed parameters can be found in each raw file under the "instrument method" section.

The mass spectrometry settings on the Eclipse mass spectrometer were set as follows: FAIMS separation was performed with the following settings: inner and outer electrode temperatures were 100 °C, FAIMS carrier gas flow was 4.4 L/min, compensation voltages (CVs) of -50, -60, and -70 V were used in a stepwise mode during the analysis. The ion transfer tube temperature was set to 275 °C. The mass spectrometer was operated in a data-dependent mode with cycle time 1 s, using the following full scan parameters: m/z range 375–1300, nominal resolution of 120,000, with a target of 100% charges for the AGC, and a maximum injection time of 100 ms. For HCD MS/MS scans, single normalized collision energy (NCE) of 32% (PhoX) and 30% (DSSO) was used for single HCD experiments and stepped HCD values of 25%, 27%, 32% for PhoX and 21%, 25%, 32% for DSSO. Precursor ions were isolated in a 1.6 Th window with no offset and accumulated for a maximum of 70 ms or until the AGC target of 500% was reached. The resolution for MS2 scans was set to 30,000. Precursors of charge states from 3+ to 6+ were scheduled for fragmentation. Previously targeted precursors were dynamically excluded from fragmentation for 15 s.

### Data analysis
Raw files were analyzed using Thermo Proteome Discoverer (v. 3.1.0.638). Searches were performed against the Cas9 sequence (Uniprot ID: Q99ZW2) plus a Crapome database (downloaded from https://www.thegpm.org/crap/). Linear peptides were identified using MS Amanda search engine (v. 3.0.20.558)[39] and the crosslinked peptides were identified using MS Annika (v. 3.0)[40–42]. The search workflow included a recalibration step for each file, followed by a first search using MS Amanda to identify linear peptides and monolinks. Subsequently, spectra with highly confident identifications of a linear peptide were filtered out and not considered for the cross-link search. Finally, a crosslink search was performed using MS Annika. The

workflow used in Proteome Discoverer is shown in Supplementary Figs. 6 and 7. Search parameters for linear and crosslink searches can be found in Supplementary Table 2. The FDR was estimated using the MS Annika validator node with 1% FDR (high confidence) for Cas9 crosslinking data on CSM and residue pair levels. The FDR calculation is based on a target-decoy approach[40]. For crosslinked peptide identification, MS Annika performs automatic spectrum de-noising using peak picking as described in the MS Amanda scoring algorithm[39]. In short, for each 100 Dalton window, the top $m$ most intense peaks are picked, where $m$ is a value between 1 and 10 (inclusive). All possible values for $m$ are tested, and the one that yields the highest peptide identification score is selected. This approach avoids the inclusion of low-intensity, potentially noisy peaks in the scoring of spectrum matches and therefore circumvents a decrease in identification scores originating from Astral spectra, which are more complex compared to spectra from the Eclipse. However, this procedure cannot be carried out during doublet detection for cleavable crosslinkers where the complexity of Astral spectra may result in more false positive doublet identifications that have to be corrected for during validation, ultimately leading to a slightly worse performance compared to non-cleavable crosslinkers.

MS1 and MS2 scan numbers, average cycle times and total run times were extracted from each raw file using RawVegetable v. 2.1.0[43].

The Cas9 model shown in Fig. 3F and Supplementary Fig. 8 was generated using AlphaFold 3[44] via the AlphaFold web service. The structure was predicted from the Cas9 amino acid sequence using a diffusion-based deep learning approach, which iteratively refines atomic coordinates to produce 3D structural models with associated confidence estimates. The resulting model was imported into UCSF ChimeraX (v.1.9)[45], and crosslinks were mapped onto the structure using XMAS[46], a ChimeraX extension that enables direct integration of results from multiple CL-MS search engines.

For data filtering and visualization Python 3.9.7 was used with the following packages: pandas (v 1.3.4)[47], numpy (v 1.20.3)[47,48], matplotlib (v 3.4.3)[49] (pyplot, venn (v 0.11.6)), seaborn (v 0.11.2)[50], scipy (v 1.7.1) and bioinfokit (v 1.0.8)[51,52].

### Computational optimizations for Astral data analysis in our cross-linking search engine MS Annika
Search and identification of crosslinks from Astral mass spectra is more challenging due to the increased size and complexity of the data. Initial searches with our cross-linking search engine MS Annika exhibited long runtimes and often failed to complete even on larger compute clusters due to the large amount of memory needed. We optimized the search in MS Annika for Astral data by benchmarking performance-critical code and ultimately implementing a more memory-efficient representation for crosslink-spectrum-matches, manually freeing memory of objects that are not needed anymore and forcing garbage collection more frequently. This led to a manyfold increase in search speed and a substantial decrease in memory consumption, which, as a result, now enables searches of Astral data to be efficiently run on standard computer hardware. These changes were implemented in MS Annika version 3.0.5, which is freely available at https://github.com/hgb-bin-proteomics/MSAnnika.

### MS1 feature matching to crosslink IDs
Raw files were converted to mzML format using ThermoRawFileParser (v1.4.5)[53]. Peptide features were extracted from the mzML files using biosaur2 software (v0.2.23)[54], with a minimum peptide feature length of one scan. Peptide features were then matched to identified CSMs using the following criteria: compensation voltage and charge state had to match, an m/z tolerance of 0.01 was applied, and the retention time difference was minimized. The retention time difference between the CSM and the peptide feature ("rt_diff"), the intensity of the monoisotopic peak at its apex ("intensityApex"), and the sum of the

monoisotopic peak intensities across all scans ("intensitySum") were added to the result files.

## Crosslink quantitation in Skyline

To quantify crosslinked peptides, Skyline (64 bit, v. 21.2.0.425, e653b4c5e)[55,56] was used, which includes native support for cross-linking workflows. Crosslinkers were defined under "Edit" following "Edit Structural Modifications", with the "Crosslinker" checkbox enabled and residue specificity set as K. Peptides were added using the "Insert Peptides" dialog, using bracket notation to specify crosslinks in the format:

```
Peptide sequence A-peptide sequence B-[cross-
linker@residue A,residue B]
E.g: DFNKVPNSIR-IVQSKSGLNMENLANHEHLLSPVR-[PhoX@4,5]
```

Skyline generates automatically theoretical fragment ions representing both cleaved (in case of cleavabel crosslinkers) and intact crosslinked products. The Cross-link Transition Calculator plugin (v1.2) was used, which automatically computs transition m/z values for both intact and fragment ions. Transition settings "Filter" were set to precursor charge 3, 4, 5, 6, ion charge 1, 2, 3 and ion type to y, b and precursor ions. Only 3 product ions for precursor quantitation were used. Full scan settings were set to 120000 resolution for Orbitrap MS1 scans and 30000 resolution for MS2 scans. Acquisition mode was set to DDA. Other settings were left on default. Raw files from either Orbitrap Astral or Orbitrap Eclipse instrument and imported into Skyline using File → Import → Results. Chromatographic peaks for each precursor were extracted[57], and peak area data were exported using Skyline's built-in reporting tools (File → Export → Report). Skyline enabled detailed visualization of crosslink identifications, with each precursor shown as a pair of modified peptides and crosslinked residues highlighted in green, bold, underlined text. Fragment ions were displayed using bracketed ion notation (e.g., y16-y7), and co-elution of fragment ions was inspected manually to validate each CSM.

## Reporting summary

Further information on research design is available in the Nature Portfolio Reporting Summary linked to this article.

## Data availability

The mass spectrometry proteomics data have been deposited to the ProteomeXchange Consortium (http://proteomecentral.proteomexchange.org) via the PRIDE partner repository[58,59] with the dataset identifier PXD059096. Source data are provided with this paper.

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

## Acknowledgements

This work was supported by the infrastructure funding 4th call 2022/01 (AT-SCP) of the Austrian Research Promotion Agency (FFG). This work was further funded by the ESPRIT program project number ESP 566 (Grant-DOI 10.55776/ESP566, F.M.), the F 8801-B Meiosis project (Grant-DOI 10.55776/F88) and project P35045-B (Grant DOI 10.55776/P35045, M.J.B.) of the Austrian Science Fund (FWF). J.B.'s work was funded by the FWF (grant DOI 10.55776/ESP497). All LC-MS/MS analyses in Vienna were performed on the Vienna BioCenter Core Facilities instrument pool. We thank the MS core facility under Elisabeth Roitinger for support and help with setting up the Orbitrap Eclipse and Astral mass spectrometer. Furthermore, we thank Manuel Matzinger and Rupert Mayer for proofreading and suggestions. This research was funded in whole, or in part, by the Austrian Science Fund (FWF). For open access, the author has applied a CC BY public copyright license to any Author Accepted Manuscript version arising from this submission.

## Author contributions

F.M. supervised and conceptualized the study, designed and performed sample processing as well as MS experiments, performed data analysis and visualization (figures, structural models, plots), writing—original draft, writing—review and editing, funding acquisition (FWF ESPRIT

ESP566); J.B. performed the MS1 feature mapping and quantitation of crosslinked precursors and wrote the corresponding method section, M.J.B. adapted MS Annika for Astral data, wrote the corresponding method sections and performed high performance computing crosslink searches; K.S. maintained and equipped the Orbitrap Eclipse and Astral instrument, corrected the manuscript and provided advice for instrument setups; V.D. and K.M. supervised the study. All authors revised and agreed on the manuscript.

## Competing interests

The authors declare no competing interests.
