## [Transparent Peer Review file · Nature Communications]

Breaking barriers in crosslinking mass spectrometry with enhanced throughput and sensitivity using Orbitrap Astral

Corresponding Author: Dr Fränze Müller

Version 0:

Reviewer comments:

Reviewer #1

(Remarks to the Author)

The authors have performed extensive additional work in response to the reviewer's original comments and this manuscript is now much improved as a result.

The majority of my comments have been addressed, and I now only one question and one general comment and I recommend the publication of this work.

Fig 2c says "Gradient length optimization using fixed parameters (500 AGC, 100 ms injection time). While protein identifications decline with longer gradients (from 30 to 70 minutes)". The figure shows 1730 HeLa protein IDs in the 70 min gradient and 2582 proteins in the 30 minute gradient. In section 2.1 you state: "To evaluate the effects of gradient length on protein identification and MS1 mass accuracy, a fixed AGC target of 500 and MS1 injection time of 100 ms were used to acquire 10 ng of HeLa lysate across 30-minute and 70-minute active chromatographic gradients. Protein identifications decreased markedly from 2,582 with the 30-minute gradient to 1,730 with the 70-minute gradient".

It seems very unusual that a longer gradient would result in fewer protein identifications as depicted in Fig. 2c and described in section 2.1.

In section 2.4 you describe what I would expect: "In shotgun proteomics, longer chromatographic separation times have been shown to improve protein identifications for injection amounts up to 1 µg." and "The same pattern was observed for general protein identification using E. coli ribosome proteins within the complex matrix, where protein identifications rose steeply from 70 to 152 between 10 and 70 minutes" which is what I would expect. This is shown in the current work in Fig. 6f where an increase in protein IDs from 70 to 152 E. coli proteins is observed going from a 10 to a 70 minute gradient.

I am surprised that Fig. 2c shows the opposite, and unexpected, trend. Do you have an explanation for this? Some explanation should probably be given.

RE the sample complexity issue, which you address in section 2.4: "However, unique crosslink identifications remained low overall, with a maximum of 8 URPs at 180 minutes, likely due to the sample complexity and the absence of any crosslink enrichment strategy." It would have been great to have a comparison of what you could see in terms of crosslinks before and after spiking the HeLa lysate, and also the difference between the 2 instruments. I agree it is hard to see crosslinks against a complex background, but your data suggests the Astral's faster scan speed and ability to detect low abundance precursors might help here. I appreciate that you added this section. I still feel more work on complex samples and on multi-protein complexes would have improved this manuscript. Perhaps you will consider this in future work.

Along the same lines, based on Fig. S9, the majority (>50%) of your crosslink identifications span distances greater than those expected based on the structure of Cas9. From the circle plots it looks like you are seeing a large fraction of all possible K-K crosslink combinations in your purified sample regardless of the K-K distance. In the complex sample using the methods presented here it was possible to detect 8 unique crosslinks. I think the value of the type of work presented here

would be even greater if it tackled challenging real world use cases: methods to improve the detection of structurally plausible crosslinks in multi-protein complexes (either purified or not). Which is the typical goal of crosslinking studies.

Reviewer #2

(Remarks to the Author)

This reviewer's previously raised concerns have been addressed.

REVIEWERS' COMMENTS

Reviewer #1 (Remarks to the Author):

The authors have performed extensive additional work in response to the reviewer's original comments and this manuscript is now much improved as a result.

The majority of my comments have been addressed, and I now only one question and one general comment and I recommend the publication of this work.

Fig 2c says "Gradient length optimization using fixed parameters (500 AGC, 100 ms injection time). While protein identifications decline with longer gradients (from 30 to 70 minutes)". The figure shows 1730 HeLa protein IDs in the 70 min gradient and 2582 proteins in the 30 minute gradient. In section 2.1 you state: "To evaluate the effects of gradient length on protein identification and MS1 mass accuracy, a fixed AGC target of 500 and MS1 injection time of 100 ms were used to acquire 10 ng of HeLa lysate across 30-minute and 70-minute active chromatographic gradients. Protein identifications decreased markedly from 2,582 with the 30-minute gradient to 1,730 with the 70-minute gradient".

It seems very unusual that a longer gradient would result in fewer protein identifications as depicted in Fig. 2c and described in section 2.1.

In section 2.4 you describe what I would expect: "In shotgun proteomics, longer chromatographic separation times have been shown to improve protein identifications for injection amounts up to 1 μ g." and "The same pattern was observed for general protein identification using *E. coli* ribosome proteins within the complex matrix, where protein identifications rose steeply from 70 to 152 between 10 and 70 minutes" which is what I would expect. This is shown in the current work in Fig. 6f where an increase in protein IDs from 70 to 152 *E. coli* proteins is observed going from a 10 to a 70 minute gradient.

I am surprised that Fig. 2c shows the opposite, and unexpected, trend. Do you have an explanation for this? Some explanation should probably be given.

We agree that this topic appears controversial. However, the effect of gradient length depends on both the injection amount and the sample type. While high injection amounts benefit from longer gradients, low injection amounts do not; in fact, they often suffer from longer gradients due to broader peaks and, consequently, lower precursor intensities. This has already been demonstrated for single-cell and other low-input samples. We hypothesize that 10 ng represents a boundary case at which longer gradients begin to lose their benefit. In our experiments, shorter gradients yielded higher peak intensities, which proved more advantageous than extending the gradient length.

As shown in the figure below, the gain in peptide identifications increases with gradient length up to approximately 70 minutes, after which the effect diminishes. It is important to note that this observation applies only to HeLa lysates. Crosslinked samples, whether enriched or not, differ substantially from standard proteomics samples—not only in their requirements for higher injection amounts to detect crosslinks, but also in their chromatographic behavior, as discussed in our manuscript. Therefore, although the influence of gradient length may seem controversial, considering the sample type is essential for meaningful optimization.

To increase the clarity of our manuscript, we have removed subfigure 2C from Figure 2 and now present gradient optimization results exclusively for the crosslinking samples.

RE the sample complexity issue, which you address in section 2.4: “However, unique crosslink identifications remained low overall, with a maximum of 8 URPs at 180 minutes, likely due to the sample complexity and the absence of any crosslink enrichment strategy.” It would have been great to have a comparison of what you could see in terms of crosslinks before and after spiking the HeLa lysate, and also the difference between the 2 instruments. I agree it is hard to see crosslinks against a complex background, but your data suggests the Astral’s faster scan speed and ability to detect low abundance precursors might help here. I appreciate that you added this section. I still feel more work on complex samples and on multi-protein complexes would have improved this manuscript. Perhaps you will consider this in future work.

Along the same lines, based on Fig. S9, the majority (>50%) of your crosslink identifications span distances greater than those expected based on the structure of Cas9. From the circle plots it looks like you are seeing a large fraction of all possible K-K crosslink combinations in your purified sample regardless of the K-K distance. In the complex sample using the methods presented here it was possible to detect 8 unique crosslinks. I think the value of the type of work presented here would be even greater if it tackled challenging real world use cases: methods to improve the detection of structurally plausible crosslinks in multi-protein complexes (either purified or not). Which is the typical goal of crosslinking studies.

Thank you again for your comments. We will certainly continue to find smart ways how to analyze complex samples in a biologically meaningful way in the future. For this manuscript we feel the ribosome sample was a good choice to show how sample complexity influences the effect of gradient length, but we agree showing a multi-protein complex would have been interesting as well. Nevertheless, such multi-protein complexes are certainly on our agenda.

Reviewer #2 (Remarks to the Author):

This reviewer's previously raised concerns have been addressed.